# The Effects of Smoking on Telomere Length, Induction of Oncogenic Stress, and Chronic Inflammatory Responses Leading to Aging

**DOI:** 10.3390/cells13110884

**Published:** 2024-05-21

**Authors:** Shreya Deb, Joseph Berei, Edward Miliavski, Muhammad J. Khan, Taylor J. Broder, Thomas A. Akurugo, Cody Lund, Sara E. Fleming, Robert Hillwig, Joseph Ross, Neelu Puri

**Affiliations:** 1Department of Biomedical Sciences, University of Illinois College of Medicine at Rockford, Rockford, IL 61107, USA; sdeb3@uic.edu (S.D.); joseph-berei@uiowa.edu (J.B.); edward.miliavski@hcahealthcare.com (E.M.); mkhan290@uic.edu (M.J.K.); trager2@uic.edu (T.J.B.); takuru2@uic.edu (T.A.A.); codyjl2@uic.edu (C.L.); 2Department of Pathology, UW Health SwedishAmerican Hospital, Rockford, IL 61107, USA; sfleming@swedishamerican.org; 3Department of Health Sciences Education, University of Illinois College of Medicine at Rockford, Rockford, IL 61107, USA; rhillwig@uic.edu; 4Department of Family and Community Medicine, University of Illinois College of Medicine at Rockford, Rockford, IL 61107, USA; joeross@uic.edu

**Keywords:** telomeres, smokers, *IFN-γ*, *hTERT*, *TRF2*, *POT1*, *ISG15*, telomere position effect, aging

## Abstract

Telomeres, potential biomarkers of aging, are known to shorten with continued cigarette smoke exposure. In order to further investigate this process and its impact on cellular stress and inflammation, we used an in vitro model with cigarette smoke extract (CSE) and observed the downregulation of telomere stabilizing *TRF2* and *POT1* genes after CSE treatment. *hTERT* is a subunit of telomerase and a well-known oncogenic marker, which is overexpressed in over 85% of cancers and may contribute to lung cancer development in smokers. We also observed an increase in *hTERT* and *ISG15* expression levels after CSE treatment, as well as increased protein levels revealed by immunohistochemical staining in smokers’ lung tissue samples compared to non-smokers. The effects of *ISG15* overexpression were further studied by quantifying *IFN-γ*, an inflammatory protein induced by *ISG15*, which showed greater upregulation in smokers compared to non-smokers. Similar changes in gene expression patterns for *TRF2*, *POT1*, *hTERT*, and *ISG15* were observed in blood and buccal swab samples from smokers compared to non-smokers. The results from this study provide insight into the mechanisms behind smoking causing telomere shortening and how this may contribute to the induction of inflammation and/or tumorigenesis, which may lead to comorbidities in smokers.

## 1. Introduction

The shortening of telomere length has been identified as a potential hallmark of cellular senescence, biological age, and various age related comorbidities, including cardiovascular diseases and cancer [1]. As part of their protective function, telomeres preserve genomic stability by preventing the fusion of chromosome ends, limiting nucleolytic decay and atypical recombination, and reducing DNA damage responses by minimizing chromosomal end exposure [2]. However, as progressive telomere shortening reaches a threshold, DNA damage-repair systems begin to recognize the unprotected chromosome ends as the DNA double-strand breaks [3,4]. Significantly shortened telomeres can activate DNA damage responses and result in cellular senescence or apoptosis [5]. Additionally, in the absence of functional tumor suppressor genes, the chromosomal instability caused by telomere shortening can contribute to tumorigenesis [6,7].

Telomeres are associated with a group of six proteins, known as the shelterin complex, which specifically recognize telomeric DNA and conform to its chromatin, forming a capped structure which aids in the genomic stability of chromosome ends [8,9]. The shelterin complex consists of *TRF1*, *TRF2*, *POT1*, *TIN2*, *TPP1*, and *Rap1* proteins [10]. Studies based on shelterin proteins have shown that *TRF1* and *TRF2* specifically bind to the double-stranded region of telomeric DNA, while *POT1* binds to the single-stranded region of telomeric DNA [11]. 

Previous studies have shown that *TRF1* plays an important role in regulating telomerase activity and protecting the integrity of telomere length [12]. *TRF1* is expressed throughout all stages of the cell cycle, and the binding of this protein to the double-stranded region of telomeric DNA inhibits the activity of telomerase [13]. A study conducted on telomerase-positive tumor cells showed that overexpression of *TRF1* led to the gradual shortening of telomeres, whereas deletion of *TRF1* led to telomere elongation [14,15]. *TRF2* plays a major role by binding to the telomere 3’ overhang, forming the t-loop and carrying it to the junction where it invades the double-stranded region [16]. Several studies have demonstrated that *TRF2* deletion results in the induction of ATM-dependent apoptosis, resulting in unprotected chromosome ends that are phosphorylated with the active form of ATM kinase [17,18,19,20,21]. When *POT1* is not functioning properly or dissociates from the telomere structure, it has been found to cause unfolding of the telomere loop and the induction of ATR-dependent DNA damage responses [22,23,24]. Another study revealed that *POT1* deletion resulted in a decrease in the interaction and activity of telomerase with telomeres [25].

Along with their critical roles in telomere maintenance, recent studies have identified several non-canonical roles played by various members of the shelterin complex. Both *TRF1* and *TRF2* have shown strong interactions with hundreds of non-telomeric sites [26,27]. *TRF1* was found to be critical for iPSC proliferation, as its inhibition reduced the efficiency of in vivo reprogramming [28,29]. Additional roles of *TRF1* in cell cycle regulation were also determined via its positive interaction with Aurora B and Nek2 [30,31]. Meanwhile, studies also reported the direct influence of *TRF2* on various cell signaling pathways, including natural killer (NK) cell activation and tumor-promoting myeloid derived suppressor cell (MDSC) recruitment [32,33]. Inhibition of *TRF2* also altered the Sox2 and c-Myc stem cell markers, while another study revealed that *TRF2* mediated vascularization via VEGF-A interaction [34,35,36]. Other proteins associated with the shelterin complex, such as Repressor activator protein 1 (*Rap1*), are implicated in several non-telomeric functions within and outside the nucleus. One study showed that *Rap1* plays a significant role in the NF-kB signaling cascade, which influences inflammatory and immune responses. In the same study, *Rap1* was also found to be involved in the signaling of other pathways, such as the signaling of the metabolism in *Rap1*-deficient mice [37]. Similarly, upregulated and downregulated expression of other shelterin complex proteins, such as TRF interacting nuclear protein (*TIN2*), are involved in various reported diseases, such as multiple myeloma and acute myeloid leukemia, respectively [38]. Overall, these studies highlight the importance of shelterin proteins in other cellular mechanisms and disease onset.

In order to bypass senescence and control telomere attrition, telomeres recruit the enzyme telomerase for elongation. Telomerase is highly expressed in germ cells and proliferating stem cells, but is found to be less active or inactive in fully differentiated cells [39]. Telomerase remains silenced for most of the human lifetime, but is reactivated when telomeres reach a critically short threshold [39]. Overactive telomerase, without a controlling factor, can lead to cell immortalization and has been found to be expressed in over 85% of human cancers, therefore being labeled as a significant marker for cancer [40]. Compared to normal cells, telomere length is shorter in the majority of cancer cells; however, telomerase reactivation maintains telomere stability and provides uncontrolled proliferative capacity to cancer cells [41,42].

The effect of cigarette smoke is known to shorten telomere length in human blood leukocytes and buccal cavity epithelial cells [43,44]. This is an important finding, as the latest estimate of tobacco users was found to be roughly 1.3 billion adults [45]. Cigarette smoke is a well-known risk factor for many diseases, including cancer, since it contains a large array of chemicals that cause heavy oxidative stress in cells by producing free radicals [46]. Out of the four nucleotide bases, guanine is known to be more sensitive to oxidative stress because of its low oxidation potential, and is more likely to be oxidized when in a sequence of multiple Gs, which form G-quadruplexes [47,48]. Therefore, the high guanine content present at the end of the repetitive telomeric nucleotide sequence TTAGGG makes telomeres more susceptible to DNA double-strand breaks caused by oxidative stress [4,49]. 

The structure of telomeres has been shown to have a silencing effect on the genes proximal to the chromosome ends through a process known as the telomere position effect (TPE) [50]. This phenomenon is caused by the spreading of telomeric heterochromatin, which alters the expression of genes that lie in the sub-telomeric region, ranging up to a couple hundred kilobases, as a function of telomere length [51,52]. In some studies, telomere shortening has proven to alter the expression levels of genes that are farther away from the ends of chromosomes, thus giving rise to the term telomere position effect over long distances (TPE-OLD) [53]. It is believed that, as telomere length shortens, the strength of the TPE decreases and there is upregulation in the expression of otherwise silenced sub-telomeric genes [54]. One study found that the gene encoding for the rate-limiting catalytic protein of telomerase, *hTERT*, had increased expression in cancer cells with short telomeres (3 kb) compared to those with long telomeres (15 kb) [53]. The study also found that cells with shortened telomeres had at least one *hTERT* allele, which was spatially separated from telomeres and developed more active histone marks and methylation changes in the promoter region [53]. On the other hand, mutations commonly observed within the *hTERT* promoter region at two main sites, C228T and C250T, were identified as a popular mechanism for *hTERT* reactivation among several cancers [55,56]. Further investigation revealed that these site-specific mutations generate new binding sites for ETS transcription factors, thereby promoting *hTERT* gene expression levels [57,58].

Located 1 Mb distal to the end of chromosome 1p, the interferon-stimulated gene (*ISG15*) is a stress response gene which has cytokine-like immuno-modulatory properties [59,60]. Due to its location in the subtelomeric region, ISG-15 has been shown to display increased expression through the TPE and progressive telomeric shortening [52,61]. Secreted ISG-15 stimulates the production of the proinflammatory cytokine interferon gamma (*IFN-γ*), whereas unconjugated ISG-15 is a chemotactic factor for neutrophils [59,62]. Abnormal *IFN-γ* levels are associated with a number of autoimmune/inflammatory diseases [63]. Increased amounts of *IFN-γ* due to telomere shortening may lead to inflammatory responses in tissues and may be related to the development of comorbidities in smokers.

Our study indicates that lung cancer cells treated with CSE show increased expression of *hTERT*, *ISG15*, and *IFN-γ*. *IFN-γ* levels increased after cells were exposed to CSE. We also found increased expression of *hTERT* and *ISG15* proteins in lung tissue samples from smokers compared to non-smokers. *TRF2* and *POT1* gene expression was downregulated after treatment with CSE. These results indicate a mechanism of telomere shortening in smokers and show that higher levels of *IFN-γ* may provide an inflammatory environment for potential oncogenesis.

## 2. Materials and Methods

### 2.1. Patient Samples

With IRB approval, patients between the ages of 45 and 75 years old were recruited for this study and categorized into two groups—smokers and non-smokers. Only the patients who had smoked for at least 15 pack-years or had no history of smoking were chosen. Blood was drawn and buccal swabs were taken by physicians at the UI Health Mile Square Health Center LP Johnson (Rockford, IL, USA) and sent to our laboratory. Similarly, lung tissue sections of smokers and non-smokers were obtained, with the approval of IRB, from the biorepository of Indiana University Melvin and Bren Simon Comprehensive Cancer Center (Indianapolis, IN, USA).

### 2.2. Cell Lines

The PC9 non-small cell lung cancer (NSCLC) adenocarcinoma cell line (Cat. No. 90071810) and the human lung fibroblast (HLF) cell line (Cat. No. AG04432) were purchased from Sigma-Aldrich (St. Louis, MO, USA) and the Coriell Institute for Medical Research (Camden, NJ, USA), respectively. The cells were cultured in incubators at 37 °C with 5% CO_2_, as per the manufacturer’s instructions.

### 2.3. Primers

Primers for mRNA expression level analysis, as well as CpG methylation pattern analysis of the promoter regions of *hTERT*, *ISG15*, *TRF2* and *POT1*, were designed and purchased from Integrated DNA Technologies (IDT, Coralville, IA, USA). *IFN-γ* primers (Cat. No. H_IFNG_3) for qPCR were purchased from KiCqStart Primers, Sigma-Aldrich (St. Louis, MO, USA). Gene sequences were identified and validated using an Ensemble and California Genome Browser (UCSC). For CpG methylation studies on *hTERT*, the promoter region (1000 bp upstream) of the target gene was bisulfite-converted using a Zymo Bisulfite Primer Seeker in order to expose the CpG sites, which helped to design the suitable primers for *hTERT*. For next-generation sequencing (NGS), CS1 and CS2 linkers were attached to the 5′ ends of each of the forward and reverse primers, respectively. Primers were purchased with a stock concentration of 100 µM and were diluted as per requirements. The primer sequences are shown in Table 1 and Table 2.

### 2.4. Antibodies for Immunohistochemistry

In order to study the expression and localization of *ISG15* in lung tissue sections of smokers and non-smokers, we conducted immunohistochemistry (IHC) experiments. A rabbit polyclonal antibody for *ISG15* (Cat. No. 15981-1-AP) was purchased from the Proteintech Group (Rosemont, IL, USA). This was used as the primary antibody for staining and it was diluted to 1:200 in TBST with a 1% bovine serum albumin (BSA) (Cat. No. A7906) purchased from Sigma-Aldrich (St. Louis, MO, USA). Biotinylated anti-rabbit IgG secondary antibody, made in goats, (Cat. No. BA-2000-1.5) was purchased from Vector Laboratories (Burlingame, CA, USA) and prepared at a dilution of 1:1000 in TBST with 10% BSA and 7.5% Normal Goat Serum (Cat. No. S-1000) obtained from Vector Laboratories (Burlingame, CA, USA).

### 2.5. Other Materials for Immunihistochemistry 

An Avidin/Biotin Blocking Kit (Cat. No. SP-2001), Normal Goat Serum Blocking Solution (Cat. No. S-1000), a VECTASTAIN Elite ABC Kit (# PK-6100), and an ImmPACT DAB Peroxidase Substrate (Cat. No. SK-4105) were all obtained from Vector Laboratories (Burlingame, CA, USA). Hematoxylin (Cat. No. H9627) was obtained from Sigma-Aldrich (St. Louis, MO, USA). Permount mounting medium (Cat. No. SP15-100) was obtained from Fisher Scientific (Pittsburgh, PA, USA). All reagents were used in accordance with the manufacturer’s instructions.

### 2.6. Preparation of Cigarette Smoke Extract (CSE)

In order to study the effects of cigarette smoke on PC9 and HLF cells, we prepared an aqueous cigarette smoke extract using the following method. The apparatus for the extraction of cigarette smoke required a 50 mL conical tube, rubber stoppers, polypropylene tubing, 1000 μL pipette tips, and a 30 mL syringe. Once assembled, the experiment was carried out inside a fume hood. A total of 10 Marlboro unfiltered cigarettes were used to prepare a 30 mL solution of CSE. Cigarettes were marked 2 cm from the end and attached to one of two polypropylene tubes, with the syringe attached to the other. The conical tube was filled with 30 mL of PBS. After lighting the cigarette, the syringe was used to draw 30 mL puffs very slowly within 4 s, which caused the smoke to bubble through the PBS. The syringe was then detached from the tubing, and the smoke from within was released onto a paper towel. The 30 mL puffs were repeated every 30 s. Burnt out cigarettes were removed using forceps and replaced with a new cigarette. Once completed, the contents of the conical tube were transferred to a new tube covered with aluminum foil, since nicotine is light sensitive. The pH of the solution was adjusted to 7.2–7.4 and passed through a 0.2 μm filter under sterile conditions. The CSE was stored at −70 °C and used as needed. An aqueous cigarette smoke extract (CSE) was prepared and an HPLC analysis was performed on the aqueous cigarette smoke extract (CSE) using HPLC with a nicotine standard in order to validate and determine the concentration of nicotine in the solution. A gradient mobile phase consisting of methanol (MeOH) and 1% acetic acid (HAc) was used. The standard and sample were diluted to 1:1 with the gradient solution. The concentration of nicotine in the CSE, as calculated from the peaks, was found to be ~1 mg/mL. 

### 2.7. Tissue Culture Techniques

PC9 and HLF cell lines were cultured in Roswell Park Memorial Institute (RPMI 1640) medium (Cat. No. SH3002701) from Thermo Fisher Scientific (Pittsburg, PA, USA), and Dulbecco’s Modified Eagle Medium (DMEM) (Cat. No. 10-013-CV) from Corning Cellgro (Corning, NY, USA), respectively. The media were supplemented with 10% (*v*/*v*) fetal bovine serum (Cat. No. S11050) from Atlanta Biologicals (Lawrenceville, GA, USA); 1% (*v*/*v*) antibiotic-antimycotic solution for PC9; 1% (*v*/*v*) penicillin-streptomycin for HLF (100 units/mL of penicillin, 100 μg/mL of streptomycin, and 0.25μg of amphotericin B/mL) (Cat. No. 15070-063); 1% (*v*/*v*) 1 mM of sodium pyruvate (Cat. No. 11360); and 1% (*v*/*v*) 1 mM of HEPES (Cat. No. 11360), which were all purchased from Life Technologies (Carlsbad, CA, USA). PC9 cells were trypsinized with 0.25% trypsin (0.53 mM of EDTA solution) (Cat. No. 25-053-CI), and HLF cells were trypsinized with 0.05% trypsin (0.53 mM of EDTA solution) (Cat. No. 25-052-CI), which were purchased from Corning Cellgro (Corning, NY, USA). Both the cell lines were treated with media supplemented with 5% CSE and cultured in 37 °C incubators with 5% CO_2_ for a period of time before being processed for respective experiments.

### 2.8. Isolation of Blood Leukocytes

The patients’ blood was processed for leukocyte isolation as soon as a sample was received. A total of 3 mL of whole blood was layered carefully on 3 mL of Histopaque-1077 solution (Cat. No. 10771), purchased from Sigma-Aldrich (St. Louis, MO, USA), in a 15 mL conical tube. The sample was centrifuged at 400× *g* for 30 min at room temperature. The supernatant/plasma was transferred to a fresh tube and stored at −70 °C for later use. The interphase containing the leukocytes was transferred to another fresh 15 mL conical tube and 10 mL of PBS was added to the sample and mixed gently by pipetting up and down. The sample was centrifuged at 250× *g* for 10 min followed by aspiration of the supernatant. The cell pellet was resuspended in 5 mL of PBS and centrifuged at 250× *g* for 10 min. The supernatant was discarded, and the cell pellet was resuspended in 5 mL of PBS. The sample was aliquoted in 1.5 mL microcentrifuge tubes and centrifuged at 250× *g* for 10 min. The supernatant was discarded, and the pellet was resuspended in 700 µL of PBS or Trizol (Cat. No. T9424), purchased from Sigma-Aldrich (St. Louis, MO, USA), for DNA or RNA isolation, respectively.

### 2.9. Handling of Buccal Swabs

Buccal swabs were collected using a Cyto-Pak CytoSoft Brush (Cat. No. CP-5B) purchased from the Medical Packaging Corporation (Camarillo, CA, USA). For DNA or RNA isolation, the brush was transferred to a 1.5 mL microcentrifuge tube containing 700 µL of PBS or Trizol (Cat. No. T9424), purchased from Sigma-Aldrich (St. Louis, MO, USA), respectively. The tube was then vortexed for 30 s, followed by removal of the brush, and used for the respective experiments.

### 2.10. Real Time PCR

Total RNA was isolated from PC9 cells, HLF cells, blood leukocytes, and buccal epithelial cells in order to study the mRNA expression levels of *hTERT*, *ISG15*, *TRF2*, *POT1,* and *IFN-γ*. Cells were lysed using 700 µL of Trizol (Cat. No. T9424) followed by 200 μL of chloroform (Cat. No. C2432), both purchased from Sigma-Aldrich (St. Louis, MO, USA). The sample was mixed thoroughly by shaking vigorously for 15 s for phase separation followed by incubation at room temperature for 10 min and centrifugation at 12,000× *g* for 15 min at 4 °C. The colorless upper aqueous phase was transferred to a fresh tube and 0.5 mL of isopropanol (Cat. No. I9516) purchased from Sigma-Aldrich (St. Louis, MO, USA) was added, followed by thorough mixing. The sample was allowed to stand at room temperature for 10 min and centrifuged at 12,000× *g* for 10 min at 4 °C. The supernatant was discarded, and the RNA pellet was resuspended in 1 mL of 75% ethanol (Cat. No. E7023) purchased from Sigma-Aldrich (St. Louis, MO, USA). The sample was vortexed and centrifuged at 7500× *g* for 5 min at 4 °C. After discarding the supernatant, the RNA pellet was allowed to air-dry for 10 min followed by resuspension in 20 μL of molecular-grade water. The RNA was quantified using the Take 3 nanodrop method and at least 200 ng was used as a template to prepare cDNA using a High-Capacity cDNA Reverse Transcription Kit (Cat. No. 43-688-14), purchased from Applied Biosystems (Foster City, CA, USA), following the manufacturer’s protocol, and using an Eppendorf Mastercycler Gradient 5331 (Germany) thermal cycler. Real-time quantitative PCR was conducted with the use of a PowerUp SYBR Green Master Mix (Cat. No. A25742), purchased from Applied Biosystems (Foster City, CA, USA), following the manufacturer’s protocol, and using an Applied Biosystems 7300 real-time PCR system. The cycle threshold (Ct) values were recorded and normalized using GAPDH as an endogenous control. Furthermore, 2(-ΔΔCt) was calculated to evaluate the fold changes in gene expression or mRNA levels in genes of interest.

### 2.11. Isolation of DNA

For the isolation of DNA, PC9 and HLF cells were trypsinized, transferred to a 1.5 mL microcentrifuge tube and centrifuged at 3000× *g* for 5 min. The supernatant was discarded and the pellet was resuspended in 700 µL of PBS. Samples of cultured cells, blood leukocytes, and/or buccal epithelial cells stored in PBS were then processed for genomic DNA isolation using a QIAamp DNA Mini Kit (Cat. No. 51304) purchased from Qiagen (Germantown, MD, USA). As per the manufacturer’s guidelines, 20 µL of proteinase K was added to a 1.5 mL microcentrifuge tube, followed by 200 µL of the sample and 200 µL of buffer AL. The tube was vortexed for 15 s and incubated in a water bath set at 56 °C for 10 min, followed by the addition of 200 µL of 100% ethanol (Cat. No. E7023) purchased from Sigma-Aldrich (St. Louis, MO, USA). The sample was briefly centrifuged and transferred to a spin column provided with the kit and centrifuged at 6000× *g* for 1 min. Filtrate was discarded and 500 µL of buffer AW1 was added followed by centrifugation at 6000× *g* for 1 min. Filtrate was discarded and 500 µL of buffer AW2 was added followed by centrifugation at 20,000× *g* for 3 min. Filtrate was discarded and the spin column was centrifuged at 20,000× *g* for 1 min. The spin column was transferred to a fresh 1.5 mL microcentrifuge tube and 30-50 µL of buffer AE was added and incubated at room temperature for 1 min followed by centrifugation at 20,000× *g* for 3 min. The DNA was then quantified using the Take 3 nanodrop method.

### 2.12. Bisulfite Conversion of DNA

The bisulfite conversion was performed on the isolated DNA using an EZ DNA Methylation-Lightning kit (Cat. No. D5030) purchased from Zymo Research (Irvine, CA, USA). In a PCR tube, 20 µL of DNA and 130 µL of lightning conversion reagent was added. The PCR tube was placed in the thermal cycler at 98 °C for 8 min, 54 °C for 60 min, and then 4 °C for storage. The sample was transferred to a 1.5 mL microcentrifuge tube and 600 µL of M-Binding buffer was added to a Zymo-Spin IC column and placed into the collection tube. The tube was centrifuged at full speed (>10,000× *g*) for 30 s and the flow-through was discarded. Then 100 µL of wash buffer was added to the column and centrifuged again for 30 s at full speed. The filtrate was discarded and 120 µL of L-desulphonation buffer was added to the column and incubated at room temperature for 12 min followed by centrifugation at full speed for 30 s. Then, 200 µL of wash buffer was added to the column and centrifuged at full speed for 1 min. The filtrate was discarded, and the column was placed in a fresh 1.5 mL microcentrifuge tube. Next, 20 µL of molecular grade water was added and the tube was centrifuged at full speed for 30 s. The eluted bisulfite-converted DNA was then used for PCR amplification.

### 2.13. PCR of Bisulfite-Converted DNA

A polymerase chain reaction (PCR) was performed on the bisulfite-converted DNA with specific primers that were designed for the target gene promoter region 1000 bp upstream of the start codon (ATG). A master mix was made using 4 µL of a 5× buffer (Cat. No. M7405) purchased from Promega (Madison, WI, USA), 0.4 µL of dNTP (Cat. No. R0191), 2.5 µL of MgCl2, 1.4 µL of GC enhancer (Cat. No. 4398848), all three of which were purchased from Thermo Fisher Scientific (Pittsburg, PA, USA), 0.3 µL of GoTaq (Cat. No. M7405) purchased from Promega (Madison, WI, USA), 0.2 µL of forward primer, 0.2 µL of reverse primer, and 6 µL of molecular grade water. The total volume in the tube was 15 µL and 5 µL of the bisulfite-converted DNA was added to prepare a total volume of 20 µL of reaction mix. The PCR was then run for the respective annealing temperatures. 

### 2.14. Agarose Gel Electrophoresis

The PCR products were prepared for the agarose gel run by adding 5 µL of molecular grade water and 5 µL of PCR products to a PCR tube along with 2 µL of 6× loading dye (Cat. No. B7025) purchased from New England Biolabs (Ipswich, MA, USA). A ladder was prepared with 9 µL of molecular grade water, 2 µL of the loading dye, and 1 µL of Low Molecular Weight DNA ladder (Cat. No. N3233) purchased from New England Biolabs (Ipswich, MA, USA). A 2% agarose gel was prepared using agarose (Cat. No. A9539) obtained from Sigma-Aldrich (St. Louis, MO, USA) and a 1× TAE buffer and run for 1 h 30 min at 100 volts. The gel was then soaked in ethidium bromide (Cat. No. 1239-45), purchased from Thermo Fisher Scientific (Pittsburg, PA, USA), for 20 to 30 min and gel imaging was performed using a ChemiDoc MP Imaging System purchased from Bio-Rad (Hercules, CA, USA).

### 2.15. Next-Generation Sequencing (NGS)

Illumina sequencing was performed on the PCR products after they were identified on the agarose gel. The UIC sequencing core received the samples and performed the NGS, and later bioinformatics analysis was performed on the samples. After the analysis, the percentage of methylation was determined at each CpG island and further statistical analysis was performed. 

### 2.16. Telomere Length Assay

The average telomere lengths of PC9 cells, HLF cells, blood leukocytes and buccal epithelial cells were determined using an Absolute Human Telomere Length Quantification qPCR Assay Kit (Cat. No. 8918) purchased from ScienCell (Carlsbad, CA, USA). 1–5 ng of DNA was used as a template and the manufacturer’s protocol was followed to conduct the assay. The assay was conducted using an Applied Biosystems 7300 real-time PCR system and the cycle threshold (Ct) values were recorded and normalized using an SCR primer set, following which 2(-ΔΔCt) was calculated. Finally, the average telomere length of the target sample was calculated by multiplying the 2(-ΔΔCt) value by the reference sample’s telomere length. Using 92 chromosome ends in one diploid cell, the average telomere length per chromosome end was also calculated (Figure 1).

### 2.17. Enzyme Linked Immunosorbent Assay (ELISA)

An ELISA was run on smoker and non-smoker plasma samples using a Human *IFN-γ* ELISA Kit (Cat. No. ELH-IFNg-1), purchased from Ray Biotech (Peachtree Corners, GA, USA), in order to determine and compare the *IFN-γ* protein levels. The manufacturer’s protocol was used to run the assay and the levels of *IFN-γ* in the plasma of smokers and non-smokers were successfully determined and compared. The assay followed the method of a sandwich ELISA. The kit encased a 96-well plate pre-coated with the primary anti-human *IFN-γ* antibody. Plasma samples were diluted to 1:1 with assay diluent A and added to the wells along with standards, and the plate was incubated for 2.5 h at room temperature with gentle shaking. The wells were washed four times with a 1× wash buffer, a 1× prepared biotinylated secondary antibody was added, and the plate was incubated for 1 h at room temperature with gentle shaking. The wells were washed four times with the 1× wash buffer, streptavidin solution was added to each well, and the plate was incubated for 45 min at room temperature with gentle shaking. The wells were washed four times with the 1× wash buffer, the TMB reagent was added to each well and the plate was incubated for 30 min at room temperature with gentle shaking. Finally, the stop solution was added to each well and absorbance was read at 450 nm immediately. The absorbance readings were used to prepare the standard curve and determine the unknown concentrations of *IFN-γ* in the plasma samples.

### 2.18. Immunihistochemistry 

The lung tissue sections from smokers and non-smokers were single stained for *ISG15* and were examined individually. Between each step, a wash step in 1× tris-buffered saline with Triton X-100 (TBST) was completed. Sections were deparaffinized in xylene for 4 min and rehydrated in decreasing concentrations of ethanol (100%, 95%, and 70%) for 1 min each followed by two TBST washes of 5 min each. Antigen retrieval was performed using Tris-EDTA (0.37 g/L) solution with Tween-20 at 95 °C for 30 min and subsequently allowed to cool at room temperature for 20 min. Endogenous peroxidase enzymes were blocked using 3% hydrogen peroxide for 5 min followed by a water quench for 3 min. Endogenous avidin and biotin were blocked using an Avidin/Biotin Blocking Kit for 15 min each and further blocking was accomplished through 60 min of incubation in 7.5% Normal Goat Serum Blocking Solution prepared in 10% BSA. The anti-*ISG15* primary antibody made in rabbit was diluted 1:200 in 1% BSA and then applied to the corresponding slides and incubated overnight at 4 °C. The following day, a biotinylated horse anti-rabbit IgG secondary antibody was diluted in 7.5% normal goat serum in 10% BSA and applied to the slides for 30 min. An avidin/biotin complex (ABC) was formed in TBST using a VECTASTAIN Elite ABC Kit, which features a horse radish peroxidase (HRP) enzyme and was applied to the slides followed by a 30 min incubation period. Color formation was then induced by the addition of an ImmPACT DAB substrate for 3 min to elicit a brown color, with the slides incubated with VECTASTAIN Elite ABC Kit. Following the color formation, slides were rinsed in tap water and counterstained with hematoxylin for 3 min. The slides were then rinsed in tap water and dehydrated in increasing concentrations of ethanol (95%, 95%, 100%, 100%) for 1 min each and xylene for 4 min. Finally, the slides were permanently mounted using Permount and a glass coverslip. 

A tissue section of skin was stained as a negative tissue control since it is known to be a negative marker of *ISG15* expression. A tumor section of a smoker with stage IV lung cancer (non-small cell lung cancer) was used as a positive tissue control and a duplicate slide was also used for negative (without primary antibody) staining as a control. Staining intensity was graded by a pathologist on a 0–3 scale (0 = no staining, 1 = weak staining, 2 = moderate staining, and 3 = strong staining). The percentage area of the tumor cells that stained positively for a particular intensity score was then recorded and multiplied with the corresponding intensity score to produce a calculated grading score. 

### 2.19. Statistical Analysis

The experiments were performed at least two to three times and quantitative analysis was performed accordingly. The quantitative data were then analyzed for statistical analysis using Student’s *t*-test with a 95% confidence interval. A *p*-value less than 0.05 was considered to be statistically significant for all experiments.

## 3. Results

### 3.1. Demographics for the Collection of Serum, Buccal Swabs, and Lung Sections of Smokers and Non-Smokers

The demographics of the 20 non-smokers and 11 smokers who provided serum and buccal samples are displayed in Appendix A. The demographics collected for the buccal and serum samples included smoking status, sex, race, and age. The age demographics were further broken down into the following classifications: 50–59 years, 60–69 years, and greater than 70 years of age. The demographics collected for the lung tissue sections included smoking status, sex, race, and age. The age demographics were further broken down into the following classifications: 40–59 years, 60–79 years, and above 80 or unknown. The demographics for the lung tissue section of non-smokers and smokers are displayed in Appendix A. 

### 3.2. Gene Expression Levels of hTERT, ISG15, TRF2, and POT1 in PC9 Cells and HLF Cells

The gene expression levels of *hTERT*, *ISG15*, *TRF2,* and *POT1* were analyzed by qPCR in PC9 cells, an NSCLC adenocarcinoma cell line, and human lung fibroblast (HLF) cells. After one week of CSE treatment, there was a 1.9- and 1.4-fold upregulation of *hTERT* and *ISG15* in PC9 cells (Figure 2A), and a 1.6- and 2.3-fold upregulation in HLF cells, respectively (Figure 2B). *TRF2* and *POT1* exhibited a 2.1- and 1.3-fold downregulation in PC9 cells (Figure 2A) and a 1.1- and 2.7-fold downregulation in HLF cells (Figure 2B) after one week of treatment, respectively. Similarly, qPCR results at the end of two weeks of CSE treatment showed a seven- and three-fold upregulation of *hTERT* and *ISG15* in PC9 cells (Figure 2A) and a 14.8- and three-fold upregulation in HLF cells (Figure 2B), respectively. PC9 cells showed a 5.0- and 5.8-fold downregulation of *TRF2* and *POT1* after 2 weeks of CSE treatment (Figure 2A). HLF samples also exhibited a similar 9.2- and 9.8-fold downregulation of both *TRF2* and *POT1* after two weeks, respectively (Figure 2B).

### 3.3. Telomere Length Modulation of PC9 Cells and HLF Cells with Telomere Length Comparison in Smokers and Non-Smokers

Telomere length assays were performed to analyze the changes in telomere lengths of PC9 and HLF cells after treatment with CSE. Results in PC9 cells showed that the telomere length per chromosome did not decline significantly after one week compared to the untreated group. However, following two weeks of CSE treatment there was a statistically significant decrease in telomere length from 1 kb before treatment to 0.44 kb by the end of the second week (Figure 3A). In HLF cells, there was a statistically significant decrease in telomere length compared to no treatment at the ends of week 1 and week 2. Telomere length reduced significantly from 0.14 kb without treatment to 0.1 kb and 0.07 kb by the ends of the first and second weeks of CSE treatment, respectively (Figure 3B). 

### 3.4. Gene Expression Comparison of hTERT, ISG15, TRF2, POT1, and Methylation Pattern Analysis of hTERT in Smokers and Non-Smokers

qPCR experiments were performed to analyze the gene expression of *hTERT*, *ISG15*, *IFN-γ*, *TRF2,* and *POT1* in the blood leukocytes of smokers and non-smokers. The results showed that the average gene expressions of *hTERT*, *ISG15,* and *IFN-γ* were 8.1-, 4.7- and 3.8-fold higher in 11 smokers compared to 20 non-smokers (Figure 4), respectively (*p* < 0.005). The results also showed that the average gene expressions of *TRF2* and *POT1* were 2.7- and 1.6-fold lower in the smokers compared to the non-smokers, respectively (*p* < 0.005). Studies using 11 smokers and 11 non-smokers showed similar results, with upregulation of *hTERT*, *ISG15*, and *IFN-γ* and downregulation of *TRF2* and *POT1*, respectively (Appendix A). 

Next-generation sequencing (NGS) technology was utilized to analyze differences in CpG site-promoter methylation in *hTERT*-promoters. The results demonstrated that in the *hTERT*-promoter region of blood leukocytes, out of the 21 hypermethylated CpG sites only -610, -600, and -483 sites showed a significant difference of 4–7% methylation between smokers and non-smokers (Appendix A). However, there were no significant methylation differences observed for *hTERT* in buccal epithelial cells.

### 3.5. Comparing Telomere Length of Smokers and Non-Smokers

Telomere length assays were performed on the blood leukocytes and buccal epithelial cells of smokers and non-smokers. The mean telomere length of smokers of 11.3 kb was shorter than that of non-smokers at 13.5 kb (*p* < 0.05) (Figure 5). Similarly, the mean telomere length in the buccal epithelial cells of smokers of 19.8 kb was shorter than for non-smokers (28.3 kb) (*p* < 0.05) (Figure 5). The procedures were repeated with another sample of 11 smokers and 11 non-smokers. The results similarly showed that the relative telomere length in smokers compared to non-smokers in blood leukocytes was 11 kb vs. 12 kb, respectively. In the same sample, the relative telomere length in buccal epithelial cells in smokers compared to non-smokers was 19.8 kb and 29 kb, respectively (Appendix A).

### 3.6. Determining the Expression of IFN-γ in the Plasma of Smokers and Non-Smokers

An enzyme linked immunosorbent assay (ELISA) was performed on plasma samples from 11 smokers and 20 non-smokers in order to determine the protein expression levels of *IFN-γ*. The assay results showed that the concentration of *IFN-γ* in the plasma of smokers was three-fold higher (298.2 pg/mL) than the concentration of *IFN-γ* in the plasma of non-smokers (96.8 pg/mL) (Figure 6). Statistical analyses were performed and demonstrated that the results were statistically significant (*p* < 0.005) according to two-tailed independent *t*-tests.

### 3.7. Expression of ISG15 in Smoker and Non-Smoker Lung Tissue

In order to study the expression and localization of *ISG15*, immunohistochemistry (IHC) was performed (Figure 7). Amongst the 22 smoker and 20 non-smoker lung sections examined, 10 smoker samples were graded as high-expression as compared to three non-smoker samples (Figure 8). Fisher’s exact test was performed comparing *ISG15* protein expression through IHC with smoking status. Our results demonstrate that protein expression levels of *ISG15* in the lungs of smokers are higher compared to non-smokers (*p* < 0.05).

### 3.8. hTERT Expression in Smokers and Non-Smokers with Lung Cancer

In order to study the expression and localization of *hTERT*, immunohistochemistry (IHC) was performed (Figure 9). Amongst the 24 smoker and 22 non-smoker lung sections examined, 23 smoker samples were graded as high-expression as compared to eight non-smoker samples (Figure 10). Fisher’s exact tests were performed comparing *hTERT* protein expression through IHC with smoking status. Our results demonstrate that protein expression levels of *hTERT* in the lungs of smokers are higher compared to non-smokers (*p* < 0.05). A statistical analysis was conducted, and the *p*-value was calculated (using Fisher’s exact test) for the distribution of high- and low-*hTERT* expression among smokers and non-smokers. The *p*-value was found to be *p* < 0.05.

## 4. Discussion

The objective of this study was to help us to gain insight into the underlying mechanisms through which cigarette smoke shortens telomeres and causes accelerated aging. Additionally, we aimed to understand how cigarette smoke induces telomere shortening and might cause stress and/or inflammation which could lead to further comorbidities.

Telomeres are highly susceptible to DNA double-strand breaks caused by oxidative stress due to the presence of a high guanine content at the end of telomeric nucleotide sequences [49,64]. Since cigarette smoke causes oxidative stress, we used cell models that were treated with an aqueous cigarette smoke extract (CSE) to analyze the effects of cigarette smoke on telomere length [46]. Our data showed that after prolonged exposure to CSE, there was a significant reduction in the telomere length of PC9 and HLF cells compared to untreated controls (Figure 3). We also demonstrated that the telomere lengths of blood leukocytes and buccal epithelial cells were significantly shorter in smokers when compared to non-smokers (Figure 5). Furthermore, the shortening of telomeres was found to be greater in the buccal epithelial cells. This may be due to the direct and prolonged exposure of localized buccal cavity epithelial cells to cigarette smoke, as compared to leukocytes isolated from systemic whole blood. Our observations were found to be consistent with previous studies which reported that the telomere lengths of smokers were significantly shorter than those of non-smokers [65,66]. Through these studies, it appears that there is an inverse relationship between cigarette smoking and telomere length and that some cell types may be affected more than others due to differing exposure levels.

Two proteins in the shelterin complex, *TRF2* and *POT1*, play a crucial role in maintaining the structural integrity of telomeres. *TRF2* helps to protect the telomere 3ʹ overhang from being recognized as a DNA double-stranded break by forming a structural cap at the ends of chromosomes to avoid DNA damage responses [16]. *TRF2* also blocks the activation sites of PARP1 and ATM kinases, both of which are responsible for triggering DNA damage responses [17,67,68,69]. On the other hand, *POT1* interacts with the single-stranded region of telomeric DNA and prevents ATR-dependent DNA damage responses and telomere loop unfolding [23,70,71,72]. Through our in vitro studies, we found that *TRF2* and *POT1* gene expression levels were significantly reduced after increasing CSE exposure (Figure 2). Similarly, we observed that the gene expression levels of *TRF2* and *POT1* in patient blood leukocyte samples were much lower in smokers compared to non-smokers (Figure 4). These data support our hypothesis, as we expected to see a positive correlation between shorter telomere lengths in smokers and decreased *TRF2*/*POT1* gene expression levels. The correlation between smoking and telomere length has also been observed in previous studies; however, it is largely unknown how cigarette smoking may affect *TRF2* and *POT1* expression levels [73,74]. Earlier studies have reported that when telomeres are severely shortened, it may degrade *TRF2* binding sites [52,75,76]. This may result in the loss of *TRF2* recruitment to telomeres, thereby causing telomere structure de-protection/instability, loss of PARP1, ATM kinase inhibition, and upregulation of γH2AX, a potential DNA damage response biomarker [17,67,68,69,77]. These processes could create a positive feedback loop leading to accelerated telomere shortening.

The change in the expression of genes can be modulated through TPE, which was first extensively described in Saccharomyces cerevisiae [50,78]. Studies conducted on mammalian cells revealed that the TPE provides a mechanism to alter gene expression which depends on the distance of the gene from the telomere ends, as well as the heterochromatin looping of the telomeres themselves [50]. Here, we report the impact of the classic TPE and TPE-OLD on the gene expression levels of *ISG15* and *hTERT*, respectively. The classic TPE is observed in genes that neighbor telomere heterochromatin loops, whereas the TPE-OLD is observed in genes that may be over several megabases distal to the chromosome ends (Figure 11) [51,52,53]. *ISG15*, a stress response gene which has cytokine-like immunomodulatory properties, was observed to show significantly increased gene expression in association with the shortening of telomeres in our in vitro CSE exposure experiments (Figure 2) [59,79]. Additionally, *ISG15* gene expression levels were higher in smokers with shorter telomere lengths compared to non-smokers (Figure 4). Our observations are consistent with previous studies reporting a direct relationship between telomere length and the reversible *ISG15* gene-altering effects of the TPE [61].

When telomeres are long, their looped structure can extend up to several megabases in length [78,80]. Therefore, genes that are usually farther away from the linearized chromosome ends, such as *hTERT*, may be silenced due to the TPE-OLD [53]. Moreover, it has been previously reported that *hTERT* is widely expressed in developing embryos, reproductive cells, and adult stem cells but is silenced in most adult human tissues [81,82,83]. However, upregulation of *hTERT* gene expression has been more widely documented in HeLa cancer cells with short telomeres (3 kb) compared to cells with long telomeres (15 kb) and it has been found to be expressed in over 85% of human cancers [53,84]. This makes telomerase a popular target for cancer treatment as it is silenced in most adult somatic cells and active in most human cancers. Additionally, we have observed that *hTERT* expression levels were significantly upregulated in cells exposed to CSE, as well as in smokers compared to non-smokers (Figure 2 and Figure 4). Through our immunohistochemistry studies, we found that the protein expression levels of *hTERT* (Figure 9) were significantly higher in the lung sections of smokers versus non-smokers (Figure 10). Although the relationship between cigarette smoking and *hTERT* expression has not been thoroughly investigated, it is known that the harmful chemicals in cigarettes result in telomere shortening, *hTERT* mutations, and heavy cellular oxidative damage [46,55,56]. Therefore, this study adds further evidence on how cigarette smoking can promote the initiation and progression of lung cancer by regulating *hTERT* gene expression as a result of telomere shortening.

*ISG15* is an important gene that is regulated by telomere length, as discussed above. It is secreted as a primary response to diverse cell stress stimuli; additionally, its secretion stimulates the production of proinflammatory cytokines such as *IFN-γ* [59,85,86,87]. High levels of *ISG15* in the body can promote a proinflammatory environment, whereas abnormal levels of *IFN-γ* are associated with several autoimmune and inflammatory diseases [63]. Through our immunohistochemistry studies, we found that the protein expression levels of *ISG15* (Figure 7) were significantly higher in the lung sections of smokers compared to non-smokers (Figure 8). Also, we observed that the protein expression levels of *IFN-γ* were significantly higher in the plasma of smokers compared to non-smokers (Figure 6). These results can be explained due to the effect of cigarette smoke on telomere length and the resulting upregulation of *ISG15* and *IFN-γ* due to the reduction of the TPE observed in smokers. Therefore, the inhibition of the TPE is a mechanism by which telomere shortening can impact human health and diseases. These results provide evidence for how changes in the telomere length and heterochromatin structure can result in increased susceptibility to inflammation-related diseases and cancer. In addition to the studies above, we also conducted experiments on the methylation patterns at CpG sites of *hTERT* (Appendix A). Our studies indicate that *hTERT* is hypermethylated in smokers. This is consistent with the majority of studies conducted on *hTERT*-promoter methylation, which widely suggest that the methylation of specific regions within *hTERT* promoter is associated with gene activation [88]. These findings suggest that *hTERT* is a unique gene and that methylation of its promoter sequence positively correlates with gene expression [89]. These studies indicate that cigarette smoke can modulate the expression of telomere-associated proteins and other proteins, such as *hTERT* and *ISG15*, that could result in comorbidities associated with smoking. Our study has thus opened up new opportunities for future studies on the role of the TPE, shelterin proteins, and their relationship with smokers and telomere shortening. It would be interesting to further study the effects of cigarette smoke using other cells, such as the bronchial epithelial cells, which are directly exposed to cigarette smoke. Additionally, it would be worthwhile to observe the effects of cigarette smoking on other telomere associated proteins, such as *TRF1*, and its effects on DNA damage response proteins such as ATM kinase.

## 5. Conclusions

This study shows the harmful effects of cigarette smoking on telomere length and the upregulation of subtelomeric genes, which exacerbates cellular stress due to inflammation and tumorigenesis [90]. We observed that cigarette smoke causes telomere shortening both in vitro and in blood and buccal cavity cells. We found the gene expression levels of telomere stabilizing shelterin proteins *TRF2* and *POT1* were lower in smokers as compared to non-smokers, suggesting a potential mechanism to explain accelerated telomere shortening in smokers. The gene expression levels of *hTERT* and *ISG15* were higher in smokers compared to non-smokers, providing evidence for altered gene expression due to the TPE and TPE-OLD, in which accelerated telomere shortening occurs due to smoking. *hTERT*, a component of telomerase, has been shown to be upregulated in several aggressive types of human cancer [91]. Upregulation of *ISG15* leads to the release of *IFN-γ,* which is associated with autoimmune and autoinflammatory disease [45,63]. This study provides evidence for possible mechanisms behind why smokers are more susceptible to inflammatory disease as well as the risk of developing cancer. 

## Figures and Tables

**Figure 1 cells-13-00884-f001:**
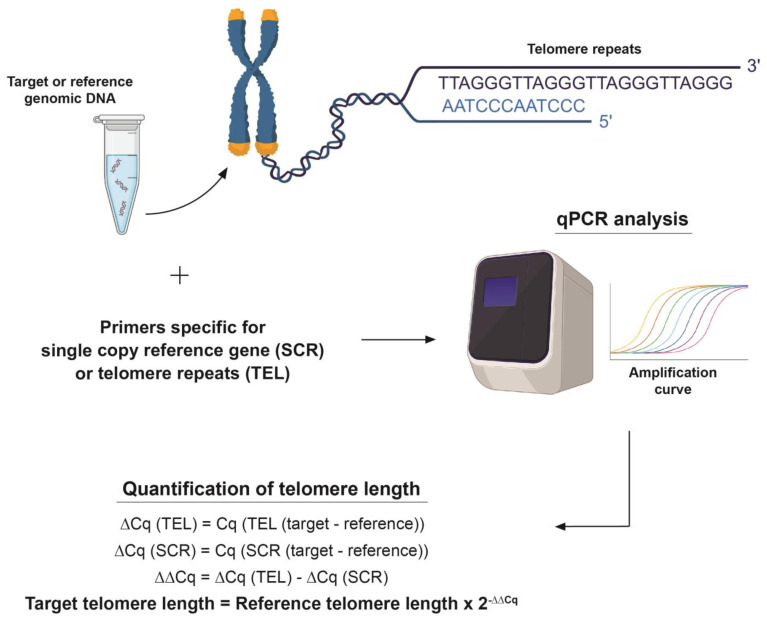
**Visual representation of TRAP assay.** The TRAP assay uses a telomere primer set which recognizes and amplifies telomere sequences, a single copy reference (SCR) primer set which recognizes and amplifies a 100 bp-long region on human chromosome #17 serves as a reference for data normalization. A reference genomic DNA sample with a known telomere length (708 ± 52 kb, per diploid cell) serves as a reference for calculating the telomere length of target samples, as shown in the figure.

**Figure 2 cells-13-00884-f002:**
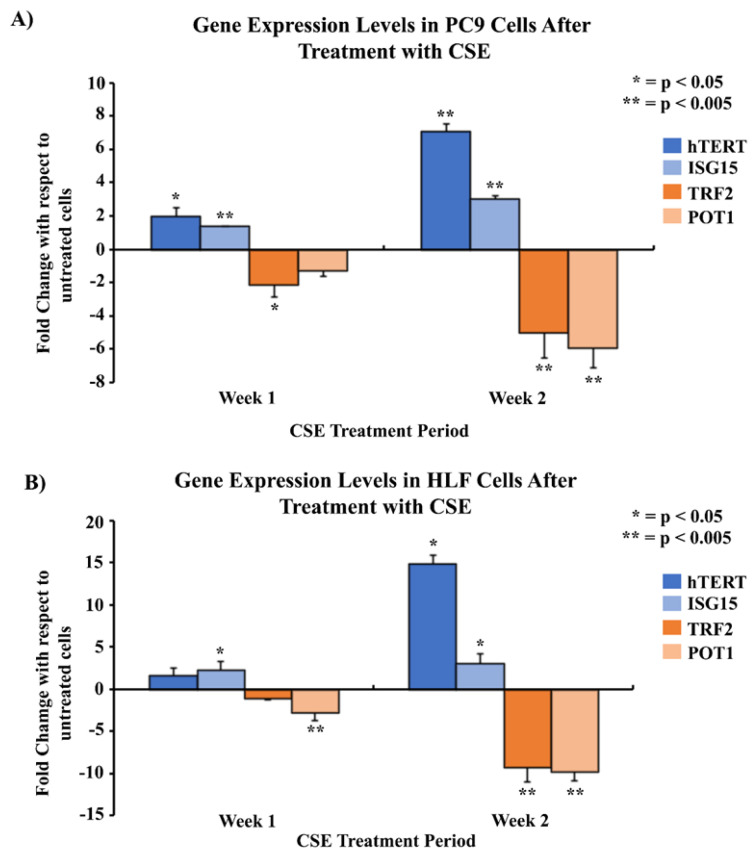
Comparison of gene expression levels of *hTERT*, *ISG15*, *TRF2,* and *POT1* in PC9 and HLF cells after treatment with CSE using qPCR. 2 × 10^5^ cells were seeded in 35 mm dishes and allowed to grow and adhere for 24–48 h, followed by treatment with media containing 5% CSE over a period of two weeks. The mRNA was quantified and analyzed using qPCR for mRNA levels of genes of interest at the end of the first and second weeks of the treatment period. Gene expression was measured in two cell lines: PC9 (**A**), and HLF (**B**). The data were normalized with GAPDH, and the results were statistically significant according to two-tailed *t*-test analyses; * = *p* < 0.05, ** = *p* < 0.005. The data were collected from triplicates (n = 3).

**Figure 3 cells-13-00884-f003:**
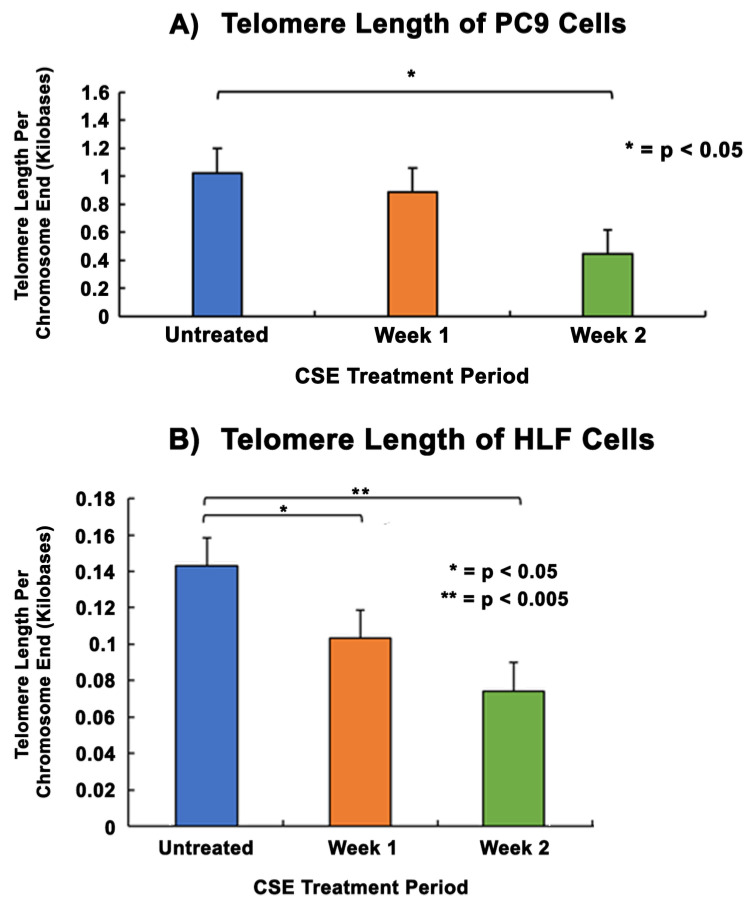
**Comparison of telomere lengths in PC9 and HLF cells after CSE treatment.** 2 × 10^5^ cells were seeded in 35 mm dishes and allowed to grow and adhere for 24–48 h, followed by treatment with media containing 5% CSE for 2 weeks. The DNA were quantified and analyzed using a telomere length assay at the end of each week of the treatment period to determine the telomere length per chromosome end. PC9 cells (**A**) and HLF cells (**B**) received two weeks of CSE treatment. The data were normalized with an assay reference primer set and the graphical representation shows the average telomere length per chromosome end of PC9 and HLF cells without CSE treatment. The data were collected from triplicates (n = 3), and the results were statistically significant according to two-tailed *t*-test analyses; * = *p* < 0.05, ** = *p* < 0.005.

**Figure 4 cells-13-00884-f004:**
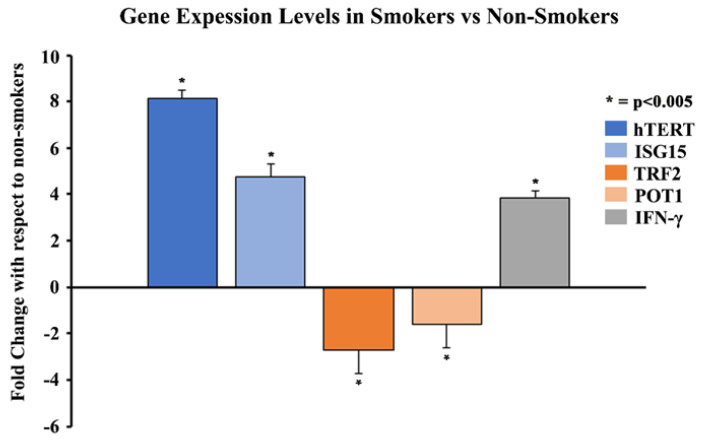
Comparison of gene expression levels of *hTERT*, *ISG15*, *IFN-γ*, *TRF2,* and *POT1* in 11 smokers vs 20 non-smokers using qPCR. Whole blood of 11 smokers and 20 non-smokers was processed to isolate leukocytes. Total RNA was extracted from blood leukocytes and was quantified and analyzed using qPCR for mRNA levels of specific genes. The data were normalized with GAPDH, and graphical representation is relative to the expression of respective genes in smokers compared to non-smokers. The data were collected from triplicates (n = 3), and the results were statistically significant by two-tailed *t*-test analysis; * = *p* < 0.005.

**Figure 5 cells-13-00884-f005:**
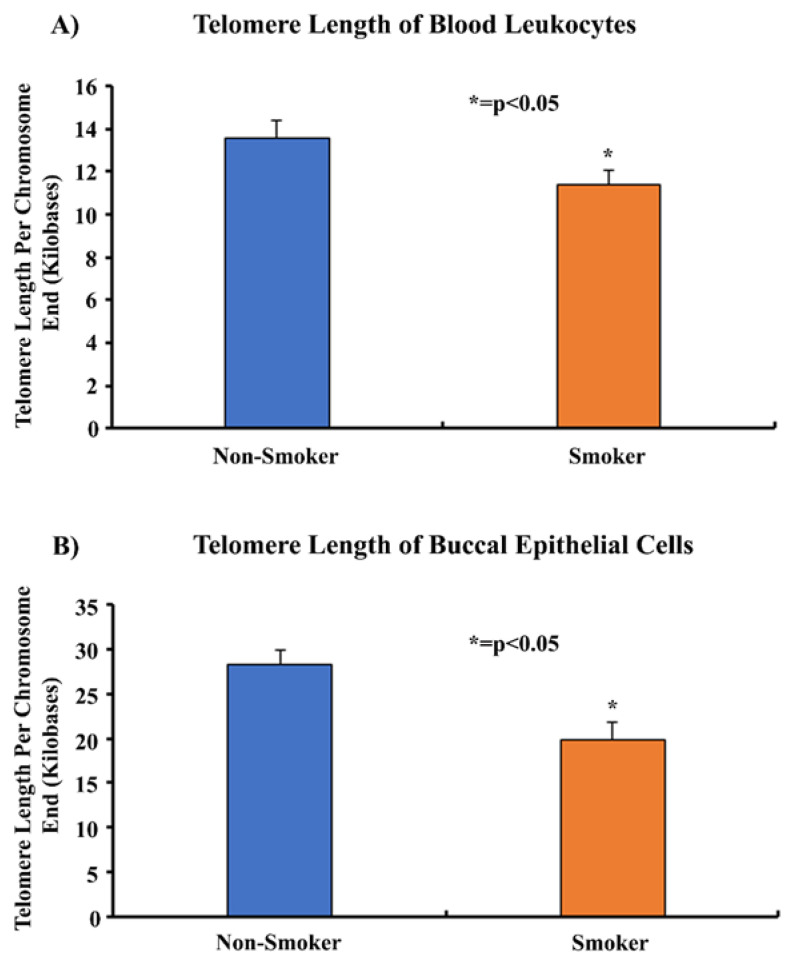
**Comparing the telomere lengths of the blood leukocytes and buccal epithelial cells of 11 smokers and 20 non-smokers.** Whole blood and buccal swabs of 11 smokers and 20 non-smokers were processed to isolate leukocytes and buccal epithelial cells. Isolated genomic DNA was quantified and analyzed in a telomere length assay to determine the average telomere length per chromosome end. The data were normalized using an assay reference primer set and were found to be statistically significant by two-tailed *t*-test analysis; * = *p* < 0.05.

**Figure 6 cells-13-00884-f006:**
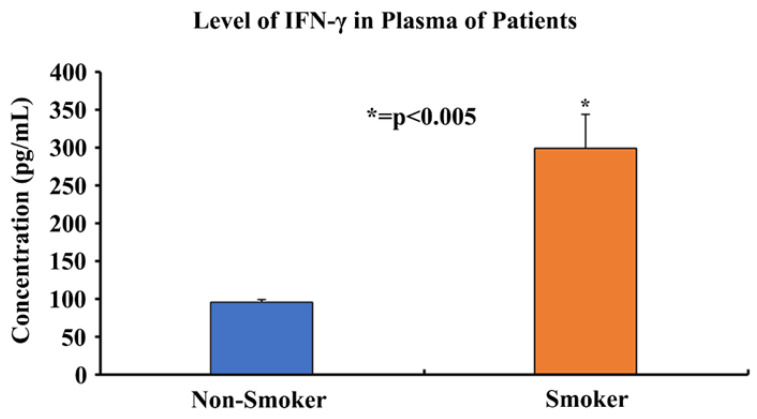
**Comparing plasma *IFN-γ* protein levels of 11 smokers and 20 non-smokers using an ELISA.** Plasma samples of 11 smokers and 20 non-smokers were processed using an ELISA to determine the *IFN-γ* protein levels. The data was found to be statistically significant by two-tailed *t*-test analysis; * = *p* < 0.005.

**Figure 7 cells-13-00884-f007:**
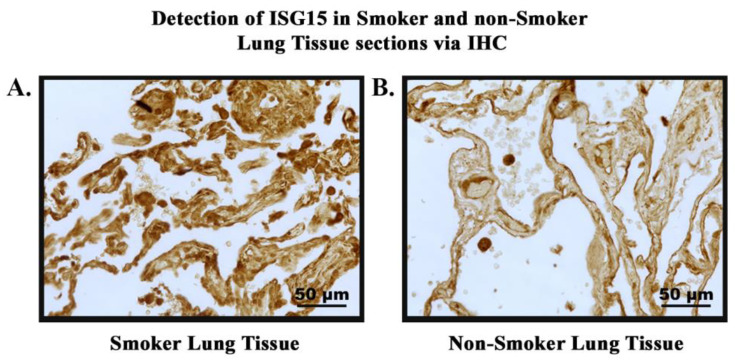
**Detection of *ISG15* in smokers’ and non-smokers’ lung tissues via IHC.** *ISG15* (brown) was detected in lung tissue sections of smokers and non-smokers. Images were taken at 40× magnification. (**A**). Normal lung tissue of smoker showing high-intensity staining of *ISG15*. (**B**). Normal lung tissue of non-smoker showing low-intensity staining of *ISG15*.

**Figure 8 cells-13-00884-f008:**
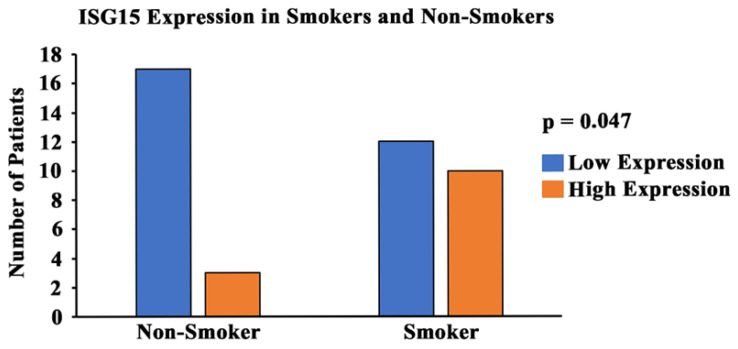
**Graphical representation of *ISG15* expression and smoking status.** A pathologist-graded *ISG15* staining intensity was conducted and a grading score was calculated between 0 and 300 (0 = no expression; 300 = high-expression). The tissues with a grading score of 250 and above were considered to be high-expression of *ISG15*, and tissues with a grading score of less than 250 were considered to be moderate/low-expression of *ISG15*. A statistical analysis was conducted and the *p*-value was calculated (using Fisher’s exact test) for the distribution of high- and low-*ISG15* expression among smokers and non-smokers. The *p*-value was found to be *p* < 0.05.

**Figure 9 cells-13-00884-f009:**
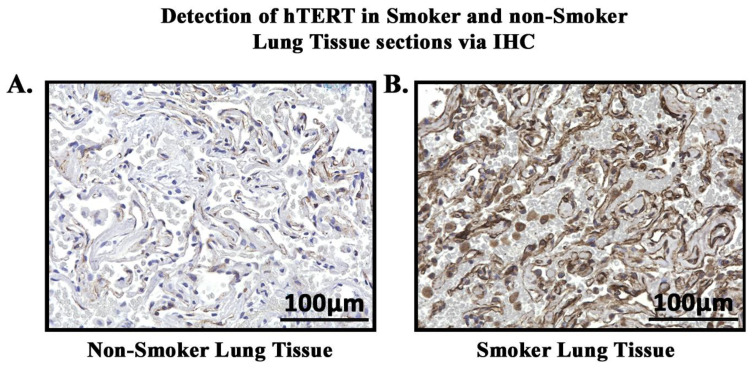
**Detection of *hTERT* in smoker and non-smoker lung tissue sections via IHC.** *hTERT* (brown) was detected in lung tissue sections of smokers and non-smokers. (**A**). Non-smoker lung tissue showing low-intensity staining of *hTERT*. (**B**). Smoker lung tissue showing high-intensity staining of *hTERT*.

**Figure 10 cells-13-00884-f010:**
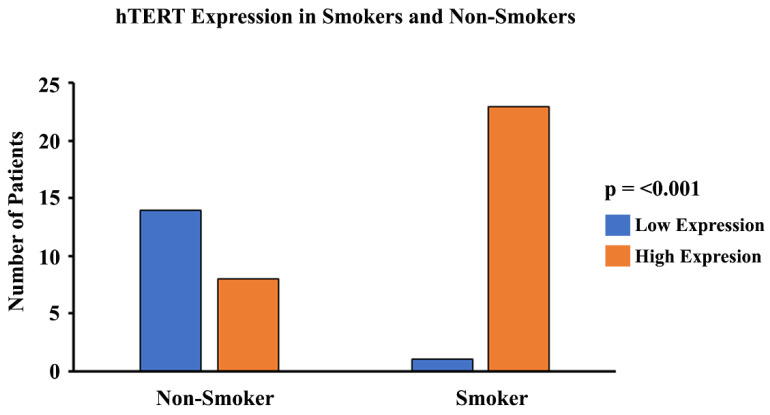
**Graphical representation of *hTERT* expression and smoking status.** A pathologist-graded *hTERT* staining intensity was conducted and a grading score was calculated between 0 and 300 (0 = no expression; 300 = high-expression). The tissues with a grading score of 150 and above were considered to be high in their expression of *hTERT*, and tissues with a grading score of less than 150 were considered to be moderate/low in their expression of *hTERT*. A statistical analysis was conducted and the *p*-value was calculated for the distribution of high- and low-*hTERT* expression among smokers and non-smokers. The *p*-value was found to be *p* < 0.001.

**Figure 11 cells-13-00884-f011:**
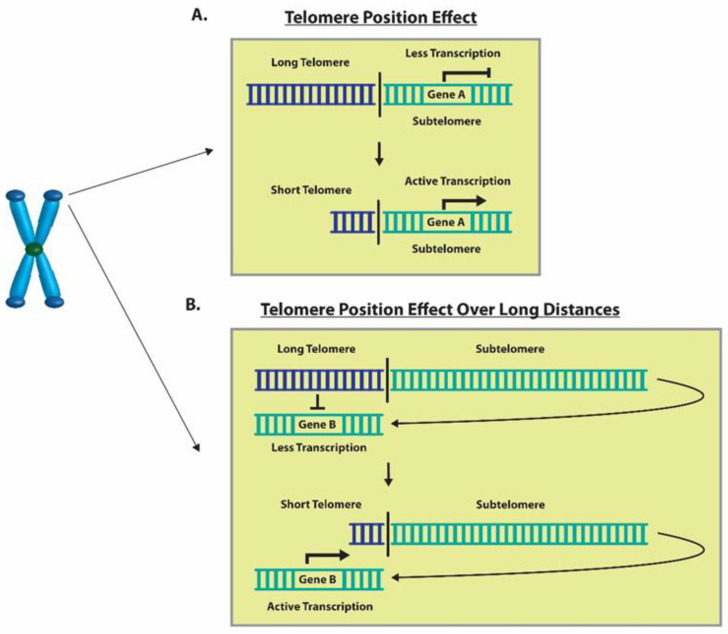
**Graphical representation of the TPE and TPE-OLD.** (**A**) An example of telomere shortening that decreases inhibition through TPE and results in increased Gene A expression. (**B**) A depiction of telomere shortening that decreases the inhibition through the TPE-OLD and results in increased gene expression of Gene B.

**Table 1 cells-13-00884-t001:** Primer sequences for qPCR.

Gene	Sequence (5′ → 3′)		DNA Bases
*hTERT*	**F:**	TGTTTCTGGATTTGCAGGTG	20
	**R:**	GTTCTTGGCTTTCAGGATGG	20
*ISG15*	**F:**	CTCTGAGCATCCTGGTGAGGAA	22
	**R:**	AAGGTCAGCCAGAACAGGTCGT	22
*TRF2*	**F:**	CTGAGTCCGCTGCCTCAAGT	20
	**R:**	ATGGTGGTTGGAGGATTCCG	20
*POT1*	**F:**	GCCACGAAGACCTGGAACTT	20
	**R:**	CCACAGAAGAAGGAATCCACG	21
*IFN-γ*	**F:**	CATTCAGATGTAGCGGATAATG	22
	**R:**	ATTCATGTCTTCCTTGATGG	20
*GAPDH*	**F:**	ATGACATCAAGAAGGTGGTG	20
	**R:**	CAGGAAATGAGCTTGACAAA	20

**Table 2 cells-13-00884-t002:** Primer sequences for CpG methylation studies.

Gene	Sequence (5′ → 3′)		DNA Bases	Annealing Temperature
*hTERT*	**F:**	GYGGGGAAGTGTTGTAGGGAGG	22	61 °C
	**R:**	CCTAATCCRAAAACCCAAAACTAC	24	
CS1	**F:**	ACACTGACGACATGGTTCTACA	22	N/A
CS2	**R:**	TACGGTAGCAGAGACTTGGTCT	22	N/A

## Data Availability

The data are contained within the present article and the Appendix A.

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
