# Peer review of "The Effects of Smoking on Telomere Length, Induction of Oncogenic Stress, and Chronic Inflammatory Responses Leading to Aging"

_cells, 2024, doi:10.3390/cells13110884_

Round 1

Reviewer 1 Report

Comments and Suggestions for Authors

Authors show the effect of cigarette smoke on telomere length. Here are the suggestions  to make the manuscript technically sound and fair in terms of literature. 

Introduction:

1) Authors introduce the roles of telomeric proteins on telomere in lines 45-50 but a vast body of evidence suggests that these proteins are also key in regulating physiology away from telomeres. Such original references (PMID: 26505215, PMID: 29515165, PMID: 31836706) and comprehensive reviews (PMID: 28264499, PMID: 33599797, PMID: 32674474,) should be cited for balanced view for the readers. 

2) Similarly authors mention the reactivation of TERT (lines 72 to 76) however do not mention that TERT reactivation occurs via promoter mutations. Cigerate smoke could be affecting how these hTERT mutations function. These statements and references such as these (PMID: 35697349, PMID: 35027447) should be cited in introduction and discussion.

Results:

1) Apart from IFN-g do the authors measure or have measured any other cytokine that is regulated by inflammation. TERT and NFkB are know to regulate each other.

2) Figure 2: can the authors show a visual display of TRAP assay

3) Figure 7: can they add error bars)

4) Figure 8. TERT antibodies are known to be non-specific. Are there any other ways to measure TERT staining. If possible please add.

5) Figure 9: please add error bars

Comments on the Quality of English Language

can improve a bit

Author Response

We would like to thank the editors and reviewers for taking the time to evaluate this manuscript. We have tried to address most of the comments to the best of our abilities. We have included a brief response to the reviewer’s comments. The additions and corrections in the manuscript have been highlighted. Please inform us if this is acceptable.

For edits, please view the manuscript draft attached.

Introduction:

Comment: Authors introduce the roles of telomeric proteins on telomere in lines 45-50 but a vast body of evidence suggests that these proteins are also key in regulating physiology away from telomeres. Such original references (PMID: 26505215, PMID: 29515165, PMID: 31836706) and comprehensive reviews (PMID: 28264499, PMID: 33599797, PMID: 32674474,) should be cited for balanced view for the readers.

Response: The role of telomeric protein in regulating physiology away from telomeres has been added and are presented in page 2, para 4.

Comment: Similarly, authors mention the reactivation of TERT (lines 72 to 76) however do not mention that TERT reactivation occurs via promoter mutations. Cigarette smoke could be affecting how these hTERT mutations function. These statements and references such as these (PMID: 35697349, PMID: 35027447) should be cited in introduction and discussion.

Response: We have described the role of mutations in the reactivation of hTERT as mentioned by the reviewer. We also describe how these hTERT mutations promote their expression in both the introduction (page 3, para 3) and discussion (page 19, para 1).

Results:

Comment: Apart from IFN-g do the authors measure or have measured any other cytokine that is regulated by inflammation. TERT and NFkB are known to regulate each other.

Response: We studied ISG15 and hTERT as they both regulate each other. ISG15 increases in cells with short telomeres and decreases following telomere elongation via hTERT [PMID: 201575433]. ISG15 also induces the production of IFN-γ which reflects the importance of ISG15 in interferon signal transduction [PMID: 16024773]. Hence we focused the study on IFN-γ. Studying NFkB is currently beyond the scope of the paper.

Comment: Figure 2: can the authors show a visual display of TRAP assay

Response: Telomere length is measured by performing a qPCR analysis. Two sets of primers were used to recognize the telomeric repeat sequence and a single copy reference (for normalization), respectively. A reference sample with known telomere length was used as control to calculate the relative telomere length of target sample. A visual display of the TRAP assay has been added to the methods section (Figure 1).

Comment: Figure 7: can they add error bars

Response: This figure is a representation of patients below/above the cut-off for IHC grading. Hence, error bars cannot be added to this graph.

Comment: Figure 8. TERT antibodies are known to be non-specific. Are there any other ways to measure TERT staining. If possible please add.

Response: Currently, there are very limited methods for staining lung sections without an antibody. The present antibody is a very well established antibody for hTERT staining (Abcam, Cat. Ab230527). We have also checked the specificity of the antibody by staining sections without the primary antibody as a control. Additionally, we stained a tissue array for specificity and found that: muscle tissue and kidney glomeruli are negative while tonsil tissue and pre-identified lung tumors are positive. These controls helped validate the specificity of the antibody. The hTERT antibody has also been verified via immunoblotting and immunofluorescence from several earlier publications [PMID: 36276465; 34702807; 33916959; 35709230; 31419512; 37732532] as well as IHC (https://doi.org/10.1101/2022.12.08.22283249).

Comment: Figure 9: please add error bars

Response: This figure is a representation of patients below/above the cut-off for IHC grading. Hence, error bars cannot be added to this graph.

Reviewer 2 Report

Comments and Suggestions for Authors

Please find below my comments.

Line 17. “Telomeres, a global biomarker of health” - I disagree with this statement. First of all, its meaning is not clear, secondly if the authors mean that telomeres length in all types of cells correlates with all pathological processes in the organism they are mistaken.

Line 18. “its negative consequences on overall health” - how the authors separate consequences caused by telomeres shorteming per se from the impact of CSE on other targets?

Names of the genes should be italicized

Line 21. “increased immunohistochemistry staining” - I suggest replacing these words with “protein levels”

Line 24. “hTERT is a subunit of telomerase and a well-known oncogenic marker. hTERT is overexpressed in over 85% of cancers and may contribute to lung cancer development in smokers”. I suggest moving these sentence to the line 20 and inserting it before the sentence starting from “We also observed an increase”

Line 29 “oncogenic stress” - please provide a definition of oncogenic stress

Line 34. “The length of telomeres has long been considered a marker of cellular aging and various aging related comorbidities including cardiovascular disease and cancer” - once again, this statement is arguable, length of the telomeres is not an universal marker of cellular ageing etc.

Line 405. In my opinion, demographics belongs to materials and methods section.

As for Telomere length assays, could you please clarify the experiment design? In my understanding, the correct experiment design is to measure telomere length at starting point of experiment in non-treated cells, next, divide cells into two groups, non-treated and treated with CSE, and compare telomere length at 1 week and 2 week timepoints in both groups. Did you use the same experiment design?

Once again, the telomere length can not be used as an universal marker of age (https://onlinelibrary.wiley.com/doi/10.1002/bies.202300187?af=R

https://academic.oup.com/biomedgerontology/article/66A/2/202/594880 and others)

Line 669. “This study shows the harmful effects of cigarette smoking on telomere length and its potential effects on the health and lifespan of individuals” - the effect on the health and lifespan has not been studied in this work. Thus, the authors can not draw such conclusion.

Author Response

We would like to thank the editors and reviewers for taking the time to evaluate this manuscript. We have tried to address most of the comments to the best of our abilities. We have included a brief response to the reviewer’s comments. The additions and corrections in the manuscript have been highlighted. Please inform us if this is acceptable.

For edits, please view the manuscript draft attached.

Comment: Line 17. “Telomeres, a global biomarker of health” - I disagree with this statement. First of all, its meaning is not clear, secondly if the authors mean that telomeres length in all types of cells correlates with all pathological processes in the organism they are mistaken.

We thank the reviewer for their critical review. As suggested by the reviewer, the language has been changed to express that telomeres are a potential biomarker of aging (abstract, line 20).   

Comment: Line 18. “…its negative consequences on overall health” - how the authors separate consequences caused by telomeres shortening per se from the impact of CSE on other targets?

As suggested by the reviewer, the wording of the sentence has been changed to “its consequences on cellular stress due to inflammation” (abstract, line 21-22).

Comment: Names of the genes should be italicized

As suggested by the reviewer, all the names of genes have been italicized.

Comment: Line 21. “increased immunohistochemistry staining” - I suggest replacing these words with “protein levels”

As pointed out by the reviewer, we have changed the sentence to read as “We also observed an increase in hTERT and ISG15 expression levels after CSE treatment as well as increased protein levels as seen by immunohistochemical staining in smoker lung tissue samples compared to non-smokers” (abstract, line 26-27).

Comment: Line 24. “hTERT is a subunit of telomerase and a well-known oncogenic marker. hTERT is overexpressed in over 85% of cancers and may contribute to lung cancer development in smokers”. I suggest moving these sentence to the line 20 and inserting it before the sentence starting from “We also observed an increase”

As recommended by the reviewer, the sentences have been moved to another location in the abstract (abstract, line 23-25).

Comment: Line 29 “oncogenic stress” - please provide a definition of oncogenic stress

As mentioned by the reviewer, the language has been altered to read as “The results from this study provide insight into the mechanism behind smoking causing telomere shortening and how this may contribute to the induction of inflammation and/or tumorigenesis which may lead to comorbidities in smokers” (abstract, line 33).  

Comment: Line 34. “The length of telomeres has long been considered a marker of cellular aging and various aging related comorbidities including cardiovascular disease and cancer” - once again, this statement is arguable, length of the telomeres is not a universal marker of cellular ageing etc. Once again, the telomere length cannot be used as a universal marker of age (https://onlinelibrary.wiley.com/doi/10.1002/bies.202300187?af=R; https://academic.oup.com/biomedgerontology/article/66A/2/202/594880 and others)

As suggested by the reviewer, the language has been adjusted to state “The shortening of telomere length has been identified as a potential hallmark of cellular senescence, biological age, and various age related comorbidities including cardiovascular diseases and cancer” (page 1, para 1, line 37-39).   

Comment: Line 405. In my opinion, demographics belongs to materials and methods section.

In our draft, we discuss the general patient samples/demographics of our study in Section 2.1 (Material and Methods) as well as Section 3.1 (Results). In both of these sections, we discuss the study cohorts included as well as the criteria needed to fit these cohorts. We believe our paper adequately provides and contextualizes the demographic data in its current format. Additionally, we give readers the option to view specific demographic data in the supplemental portion of our paper. In its present form, we feel that the demographic tables are better suited in the supplementary section.

Comment: As for Telomere length assays, could you please clarify the experiment design? In my understanding, the correct experiment design is to measure telomere length at starting point of experiment in non-treated cells, next, divide cells into two groups, non-treated and treated with CSE, and compare telomere length at 1 week and 2 week timepoints in both groups. Did you use the same experiment design?

Due to technical difficulties and the large amount of starting materials required, the telomere length was measured only after the end of treatment period between the treated vs control groups which showed very significant differences. The telomere length of PC9 cells (cancer cells) do not change significantly in two weeks as telomerase maintains telomere length in cancer cells [PMID: 30709063]. For HLF cells, since they are human fetal fibroblasts, they can be grown for several passages (more than 22) and retain high proliferative capacity [PMID: 15710616; 1638685]. They may not show significant telomere length changes at 2 weeks after 2 passages in the cells which we obtained from Coriell at passage 3. Fetal fibroblasts are routinely used for studies until passage 14-15 and can grow up to 34 passages. [PMID: 2412721; 1376315](https://e-peptide.com/peptide-shop/lingual-cytomaxes/peptide-therapy/epithalon-10mg-geroprotector-with-anti-cancer-effect-detail). Besides the CSE supplementation, all other parameters were kept the same for both groups.

Comment: Line 669. “This study shows the harmful effects of cigarette smoking on telomere length and its potential effects on the health and lifespan of individuals” - the effect on the health and lifespan has not been studied in this work. Thus, the authors cannot draw such a conclusion.

As suggested by the reviewer, we have modified the statement to read as follows: “This study shows the harmful effects of cigarette smoking on telomere length and upregulation of subtelomeric genes which exacerbates cellular stress due to inflammation and tumorigenesis” in the conclusion (page 20, para 2, line 703-705).

Round 2

Reviewer 2 Report

Comments and Suggestions for Authors

The authors have addressed all my comments and the manuscript can be published in its current state.